# Wild Wheat Rhizosphere-Associated Plant Growth-Promoting Bacteria Exudates: Effect on Root Development in Modern Wheat and Composition

**DOI:** 10.3390/ijms232315248

**Published:** 2022-12-03

**Authors:** Houssein Zhour, Fabrice Bray, Israa Dandache, Guillaume Marti, Stéphanie Flament, Amélie Perez, Maëlle Lis, Llorenç Cabrera-Bosquet, Thibaut Perez, Cécile Fizames, Ezekiel Baudoin, Ikram Madani, Loubna El Zein, Anne-Aliénor Véry, Christian Rolando, Hervé Sentenac, Ali Chokr, Jean-Benoît Peltier

**Affiliations:** 1UMR IPSiM, Université de Montpellier, Institut Agro, CNRS, INRAE, 2 Place Pierre Viala, CEDEX 2, 34060 Montpellier, France; 2Research Laboratory of Microbiology, Department of Life & Earth Sciences, Hadat Campus, Faculty of Sciences I, Lebanese University, Beirut 1302, Lebanon; 3Platform of Research and Analysis in Environmental Sciences (PRASE), Doctoral School of Sciences and Technologies, Hadat Campus, Lebanese University, Beirut 1302, Lebanon; 4USR 3290, MSAP, Miniaturisation pour la Synthèse l’Analyse et la Protéomique, CNRS, Université de Lille, 59000 Lille, France; 5Metatoul-AgromiX Platform, LRSV, Université de Toulouse, CNRS, UT3, INP, 31030 Toulouse, France; 6MetaboHUB-MetaToul, National Infrastructure of Metabolomics and Fluxomics, 31077 Toulouse, France; 7LEPSE, Université de Montpellier, Institut Agro, INRAE 2 place Pierre Viala, CEDEX 2, 34060 Montpellier, France; 8LSTM, Université de Montpellier, Institut Agro, IRD, CIRAD, INRAE, 34730 Montpellier, France

**Keywords:** PGPR, diazotroph, bacterial exudate, metabolomics, proteomics, durum wheat, plant–microbiome communication

## Abstract

Diazotrophic bacteria isolated from the rhizosphere of a wild wheat ancestor, grown from its refuge area in the Fertile Crescent, were found to be efficient Plant Growth-Promoting Rhizobacteria (PGPR), upon interaction with an elite wheat cultivar. In nitrogen-starved plants, they increased the amount of nitrogen in the seed crop (per plant) by about twofold. A bacterial growth medium was developed to investigate the effects of bacterial exudates on root development in the elite cultivar, and to analyze the exo-metabolomes and exo-proteomes. Altered root development was observed, with distinct responses depending on the strain, for instance, with respect to root hair development. A first conclusion from these results is that the ability of wheat to establish effective beneficial interactions with PGPRs does not appear to have undergone systematic deep reprogramming during domestication. Exo-metabolome analysis revealed a complex set of secondary metabolites, including nutrient ion chelators, cyclopeptides that could act as phytohormone mimetics, and quorum sensing molecules having inter-kingdom signaling properties. The exo-proteome-comprised strain-specific enzymes, and structural proteins belonging to outer-membrane vesicles, are likely to sequester metabolites in their lumen. Thus, the methodological processes we have developed to collect and analyze bacterial exudates have revealed that PGPRs constitutively exude a highly complex set of metabolites; this is likely to allow numerous mechanisms to simultaneously contribute to plant growth promotion, and thereby to also broaden the spectra of plant genotypes (species and accessions/cultivars) with which beneficial interactions can occur.

## 1. Introduction

Often rightly decried for the pollution and erosion of biodiversity it generates, the classic model of industrialized agriculture, based on unbridled use of inputs, is now widely considered as unsustainable. The evolution towards alternative models aiming to maintain high yields while respecting the environment implies not only profound changes in mentality, but also the implementation of solutions that require global rethinking of the system; these solutions include the restoration of soil fertility (through better management of organic inputs) and biodiversity of agro-ecosystems, which requires time and the acquisition of new knowledge. Evidence has been obtained that the development of Nature-Based Solutions (NBS) can efficiently contribute to a strong reduction in agricultural inputs and pesticide use [1].

The use of PGPR as biofertilizers is seen as a way to reduce the present dependence on hazardous agrochemicals. PGPRs can promote plant, water, and mineral nutrition, plant health, and ultimately plant growth, through many different processes [2]. These processes include: atmospheric nitrogen fixation; solubilization of hard-to-access nutrients; modification of root system development, including stimulation of root hair production and elongation through the secretion of active compounds such as phytohormones or phytohormone mimetics [3]; induction of plant genes involved in nutrient acquisition or plant defense [4,5,6,7]; competition with soil pathogenic microorganisms for rhizosphere resources; and exudation of phytoalexins and compounds with antibiotic activities [8,9].

A meta-analysis based on 59 published papers to evaluate the overall contribution of *Azospirillum* strains, well-known PGPRs, on growth and yield of wheat, registered an increase in aerial dry weight and seed weight of about 18% and 9%, respectively [10]. In rice, in another type of approach, *Pseudomonas* PGPR strains allowed reduction of nitrogen fertilization by 25%, without affecting crop yield [11]. Similar observations were also reported in rapeseed using *Azotobacter* as a PGPR [12]. In maize, a yield increase of about 34% was observed after inoculation with a strain of *Herbaspirillum seropedicae*, which was found to supply 37% of the plant nitrogen content through biological nitrogen fixation [13].

The amplitude of growth promotion greatly varies according to the plant-PGPR duo, and both the plant and bacterial genotypes govern the specificity and efficiency of the interaction [14,15,16,17]. The molecular mechanisms that determine the levels of specificity and efficiency are still poorly known but are likely to be multiple and very complex. They can be classified chronologically as taking part in three successive sets of processes, which condition the effectiveness of the interaction and the promotion of plant growth [14]: (i) exchange of information between the two partners, signaling their presence and bacterial chemotaxis towards the roots; (ii) colonization of the roots by the bacterial partner, based on molecular interactions between plant and bacterial surface components; (iii) effective functioning of the associative symbiosis, which requires exudation of molecules that promote plant growth by the bacterial partner colonizing the root surface [14].

An important issue regarding this chronological model is that plant growth promotion does not always appear to be strictly dependent on root colonization. First, root colonization by a given PGPR strain does not systematically result in plant growth promotion [16]. Furthermore, PGPR species can produce volatile organic compounds (VOCs) that can diffuse within the soil, through both air-filled and water-filled pores, and thereby reach host roots and promote plant growth. For instance, *Bacillus* strains can produce VOCs such as 2,3-butanediol and acetoin, which trigger growth promotion in Arabidopsis through the activation of the cytokinin pathway. Exogenous application of these two VOCs showed a similar effect to bacterial inoculation, on plant growth, while mutant *Bacillus* strains, impaired in the synthesis of these VOCs, were devoid of the ability to promote plant growth [18]. On the other hand, *Bacillus* VOCs have also been shown to modulate root-system architecture in Arabidopsis [19], and 2,3-butanediol has been shown to also induce ISR (induced-systemic resistance) through an ethylene-activation pathway [20].

With respect to crops, there is another important issue regarding the consequences of domestication and selection on the plant ability to engage in symbiotic interactions with PGPR. In a meta-analysis, the composition of the root-associated microbiome was compared between domesticated and wild ancestor species in different plants (Arabidopsis, lettuce, sugar beet, barley, sunflower and common bean) and significant differences in bacterial species abundance were observed between wild and modern varieties [21]. Likewise, a comparison of 199 ancient and modern wheat accessions, for their ability to interact with the model PGPR strain *Pseudomonas kilonensis* F113, has revealed that root colonization was lower in modern wheat, suggesting that recent wheat breeding strategies have affected PGPR symbiotic interactions in modern wheat cultivars [22]. This hypothesis was also proposed based on the observation that root microbiota diversity appears to be reduced in elite wheat cultivars, compared to ancestral forms. This led to the suggestion that introgression with ancestral species could restore the original diversity [23,24].

Wheat is an important staple crop, originating in the Middle East, and domesticated in the Fertile Crescent. The ancestor of wild wheat (*Triticum turgidum dicoccoides*) can still be found in the refuge areas of the Fertile Crescent. Studies on the microbiota of wild plants can provide information on the evolution of the relationship between wheat and its microbiota during domestication, as well as beneficial bacteria for modern crops [25].

At the molecular level, the selectivity and functioning of the symbiotic interactions are essentially dependent on exudation activities of both partners, since root exudates have to, at least, attract and feed the bacterial partner, and bacterial exudates have to trigger or mediate processes that lead to improved plant mineral nutrition and health. PGPR exudation has not yet received as much attention as root exudation [26,27,28]. Most exo-omics studies so far in PGPR have been focused on VOCs, revealing important roles played by such compounds in communication and interactions between soil microorganisms [29], as well as in plant health and growth promotion [30,31,32,33].

Apart from the information available on VOCs, the current knowledge on exo-metabolomes and exo-proteomes of PGPR species is still poor. More detailed “exo-omics” analyses are available for pathogenic bacterial species. For instance, in the plant pathogen *Xylella fastidiosa*, exo-proteomics and metabolomics analysis allowed the identification of eight proteases, alongside a bunch of outer-membrane proteins, periplasmic proteins and adhesins. Metabolites identified with a confidence score (>0.5) were mainly fatty acids [34].

Here, two diazotrophic PGPR strains, isolated from the rhizosphere of a wild wheat ancestor, identified in a Fertile Crescent refuge area, are characterized and behave as effective PGPRs with an elite wheat cultivar. Bacteria-free supernatants obtained after filtration of cultures of these two diazotrophic PGPR strains are shown to affect wheat root development. Exo-metabolomics and exo-proteomics analyses have revealed constitutive exudation, in absence of a plant host, of a large diversity of compounds and peptides that may play a role in plant growth promotion, such as cyclic dipeptides mimicking phytohormones, proteases and antibiotics.

## 2. Results

### 2.1. Diazotrophic Bacteria: Origin and Characterization

Sixteen diazotrophic bacterial strains were selected using NFb medium, a standard medium free from assimilable nitrogen, from the rhizosphere of a wheat ancestor (wild emmer, *T. t. dicoccoides* accession Ttd-NC-2019) selected in a non-cultivated refuge area in Lebanon. Using the same procedure of selection on semi-solid NFb medium, another sixteen rhizospheric diazotrophic bacteria were selected from the rhizosphere of the wild emmer selected in Lebanon, but grown in a French soil with chemical and physical features close to those of the initial Lebanese soil (Appendix A).

Two strains were selected from this bacterial collection, one from Lebanon and one from France. Whole genome sequencing and sequence analyses (Appendix A) revealed that the strain from Lebanon was a *Pseudomonas* species closely related to *P. urmiensis*, an Iranian strain isolated from Urmia. The other strain was identified as an *Enterobacter* species close to *E. ludwigii*. Many genes are not annotated in *P. urmiensis* and we found two unpaired contigs, suggesting differences between our strain and the reference strain present in the NCBI database, and/or incomplete sequencing of the reference strain. In this report, these two selected strains are named *BPMP-PU-28* (or “Pseu” in figures) and *BPMP-EL-40* (or “Enter” in figures), respectively.

The ability of these two bacterial strains to fix nitrogen was then evaluated using ^15^N_2_ incorporation assays. *E. coli* was used as a negative control. The data shown in Figure 1 indicated that the two selected strains were actually able to fix nitrogen, as expected from their selection on NFb medium. Further biochemical characterization revealed that *BPMP-PU-28* and *BPMP-EL-40* strains differed in functional properties that may contribute to PGPR activity. The capacity to produce auxin (indole-3-acetic acid, IAA) in presence of tryptophan was found to be about five times larger in the latter species (Appendix A). In contrast, production of cyanide (which may contribute to defense against pathogens) was observed only in the former species (Appendix A). *BPMP-PU-28* also appeared more efficient in solubilizing and using hardly soluble sources of phosphate or K^+^ (tri-calcium phosphate, phytate and feldspar; Appendix A).

### 2.2. PGPR Effect on a Wheat Elite Cultivar

The PGPR ability of the two selected strains was evaluated on a widely used durum wheat elite/modern cultivar (cv. Anvergur) (56% of the total durum wheat cultivated area in France in recent years: https://www.arvalis-infos.fr/_plugins/WMS_BO_Gallery/page/getElementStream.jspz?id=72353&prop=file) (accessed on 22 July 2022). Plants were grown on an artificial substrate (sterile peat/sand/vermiculite mixture; 1:1:1, *v*/*v*/*v*), ensuring that nutrients for plant growth were primarily provided by the watering solution. Plants were either inoculated or not, with *BPMP-PU-28* or *BPMP-EL-40*. The nutrient solution for the inoculated plants contained either 100 µM or 250 µM NO_3_ as the sole source of assimilable nitrogen. Uninoculated plants (controls) were watered with a solution containing either 11 mM of assimilable nitrogen (10 mM NO_3_^−^ and 1 mM NH_4_^+^; complete control nutrient solution; positive control), or such as for the inoculated plants, 100 µM or 250 µM NO_3_^−^ (negative controls).

When the watering solution contained 100 µM NO_3_^−^, no statistically significant difference, in terms of leaf surface area (LSA) recorded on a phenotyping platform (see experimental procedure), was observed between the inoculated plants and the uninoculated ones (Figure 2A). The plants were chlorotic and died 6 to 7 weeks after germination, revealing that below a certain amount of nitrogen in the medium in our experimental conditions, the plants could not survive even when inoculated with diazotrophic bacteria. In contrast, when watered with 250 µM NO_3_^−^, the plants stayed alive and completed a full life cycle with production of grains. Phenotyping revealed significant differences between the inoculated and uninoculated plants (watered with the same nutrient solution: 250 µM NO_3_^−^) in terms of leaf surface area and leaf sheath height at 5 weeks (Figure 2B,C); spikelet number per main shoot at 11 weeks (Figure 2D); thousand-seed mass and total seed mass per pot (Figure 2E,F); seed nitrogen content, and total seed nitrogen content per pot (four plants per pot) (Figure 2G–I). The growth and yield parameters were always significantly higher in the inoculated plants than in the uninoculated ones, indicating that the bacteria actually promoted plant adaptation to the low availability of nitrogen in the nutrient solution. The largest beneficial effects on plant development were often observed with *BPMP-PU-28*.

### 2.3. Development of Culture Media Suitable for Metabolomics and Proteomics Analyses of Bacterial Exudates

In the absence of information on an optimized medium for bacterial metabolomics in the literature, experiments were undertaken to develop a “minimal” medium allowing for analyses of the exo-metabolome and exo-proteome (globally named “exudates”) of diazotrophic bacteria. An objective was also to develop a bacterial growth medium compatible with, and based on, conventional plant nutrient solution, in order to make it possible to study the exudation-based dialogue between bacteria and their root partners. Compounds with a molecular weight greater than 100 Da have to be excluded from such a bacterial minimal medium, as they would interfere with metabolomics analyses. Similarly, the bacterial minimal medium has to be free from amino acids, as these molecules are likely to be secreted by the bacteria, or during symbiotic interaction by the host plant. The bacterial culture medium we developed uses a Hoagland plant medium [35], which contains only inorganic salts, as a background, without addition of yeast extract, vitamins or amino acids. Different molecules were tested as carbon sources in this Hoagland background by following bacterial growth kinetics. Lactate (final concentration: 2%, *w*/*v*, i.e., about 220 mM) was finally selected because this substrate was found to allow most of the tested diazotrophic strains, including *BPMP-PU-28 and BPMP-EL-40*, to grow quite rapidly. Typical growth curves of these two bacterial strains in this medium are shown in Appendix A. Aliquots of the growth medium were taken during the exponential and stationary phases for lactate assays (Appendix A) and pH measurements (Appendix A). The concentration of lactate in the aliquots collected during the stationary phase was still high (about 40% of the initial amount, i.e., about 90 mM), suggesting that availability of the carbon source was not the limiting factor for bacterial growth at this time. The pH recordings indicated that, with lactate as the carbon source, both bacteria strongly alkalized the medium to a pH of about 8.5 during the stationary phase. In some experiments, the buffering capacity of the growth medium was increased by adding phosphate buffer (150 mM KH_2_PO_4_/K_2_HPO_4_, pH 6.8), which kept the pH of the medium close to the initial value (6.8) throughout the culture.

### 2.4. Bacterial Exudates Affect Root Development

Experiments were carried out to investigate the effects of the exudates of the two selected diazotrophic bacterial strains on wheat growth, in hydroponics on Hoagland solution. The bacterial culture medium was buffered with 150 mM KH_2_PO_4_/K_2_HPO_4_ pH 6.8. The cultures were collected during the stationary phase, after 48 h of growth for *BPMP-PU-28* and 40 h for *BPMP-EL-40.* Cultures were centrifuged and supernatants were filter sterilized (0.22 µm). The aliquots of the sterile supernatant thereby obtained were diluted (5%) into Hoagland solution used for wheat growth. As a control treatment, wheat growth solution without added bacterial culture medium was directly supplemented with lactate and KH_2_PO_4_/K_2_HPO_4_ at concentrations similar to those that would have resulted from the addition (at 5%) of filtrated supernatant (final concentrations close to 4.5 mM and 7 mM, respectively).

When compared with the control treatment, addition of bacterial growth supernatant to the wheat growth solution was without any significant effect on shoot biomass production (Figure 3A), but significantly affected root growth and development within seven days. Furthermore, the culture supernatants of the two bacterial strains were found to have differentiated effects. Root system biomass was not affected by *BPMP-PU-28* culture supernatant, however it was increased by *BPMP-EL-40* culture supernatant (Figure 3B). Both bacterial exudates decreased total root length (Figure 3C) but increased the root diameter (strongest effect observed for the *BPMP-EL-40* culture supernatant; (Figure 3D)). Furthermore, a striking effect of *BPMP-PU-28* culture supernatant was observed on root hair density and length in root apical regions (Figure 4). Indeed, compared to the control plants and to the plants that had received *BPMP-EL-40* exudates, the plants that had received *BPMP-PU-28* exudates developed very long root hairs, which switched the root phenotype to a dauciform aspect [36], as shown in Figure 4B.

Altogether, these results provide evidence that compounds impacting wheat root system development were present in the *BPMP-PU-28* and *BPMP-EL-40* growth supernatants.

### 2.5. Comparative Metabolomics Profiling of BPMP-PU-28 and BPMP-EL-40 Exudates

Untargeted metabolomics by UHPLC-MS/MS fragmentation was applied to identify primary and secondary metabolites present in the growth supernatants of *BPMP-PU-28* and *BPMP-EL-40*. Each bacterial strain was grown in minimal buffered (150 mM phosphate buffer, pH 6.8) or unbuffered medium, and aliquots of culture supernatants were collected for metabolomics analyses during the exponential and stationary phases of the culture (at the times indicated by the arrows in Appendix A). Only metabolites identified with a score equal to or above 6.5, in at least one condition, were considered after the separation step, processing, cleaning, and peak annotation [37,38,39].

This resulted in a list of about 108 compounds, of which about 25% did not correspond to known structures and 75% were known molecules (Figure 5, Appendix A). Fatty acids and carboxylic acids together represent almost 20% of the identified compounds, and sugars represent about 5%. Some of these compounds may be involved in cell wall biosynthesis. While amino acids were not found in the bacterial growth supernatants, modified di- and tripeptides and cyclic peptides represented ca. 6.5% of the identified metabolites. It should be noted that a large percentage of metabolites (93%) are secondary metabolites, such as alkaloids and sesquiterpenoids. Overall, 18% of the metabolites identified in this study correspond to compounds that can be expected to have antibiotic effects (antimicrobial or nemato/entomopathogenic activity). For instance, columbianetin (metabolite number 528 in Appendix A), a furanocoumarin, acts as a phytoalexin and antifungal molecule [40]. It is also worth noting that some of the detected metabolites could play a role in nutrient ion acquisition by the bacteria (and the associated plant roots), such as coproporphyrin III (metabolite number 59 in Figure 5, Appendix A), which can display zincophore (Zn^2+^) or chalkophore (Cu^2+^) activity [41,42]. Moreover, N-tetradecenoyl-L-homoserine lactone (TDHL) was found in *BPMP-PU-28* and *BPMP-EL-40* exudates (Figure 5, Appendix A; compound 341). TDHL is a quorum-sensing signaling molecule modulating cell density. It is also used as a signaling molecule in bacterial interactions with higher organisms [43].

Principal components analysis (PCA) was performed to obtain an unsupervised overview of the metabolic profiles of the different *BPMP-PU-28* and *BPMP-EL-40* exudates, produced in either buffered or unbuffered bacterial growth medium, and sampled during the exponential or the stationary phases (Appendix A). In multivariate analysis, this procedure groups the samples with similar chemical profiles together. The PCA displayed a clustering of the different replicates corresponding to the same bacterial species and the same experimental conditions (buffered or unbuffered medium, sampled during the exponential or stationary phase) (Appendix A). Altogether, these results provide evidence of the reproducibility of the experiments and the robustness of the data acquisition. Overall, the PCA analysis revealed that the bacterial species was the main driving component (PC1: 35.2% of total variance) to explain the variance, as expected, but also with significant contributions from the buffered/unbuffered nature of the culture medium condition (PC2: 17%) and the phase (exponential/stationary) of the culture (PC3: 12%) (Appendix A).

Heat map analysis was used to investigate the distribution of the most discriminating metabolites between the different bacterial exudates i.e., produced in buffered or unbuffered culture media and collected at exponential or stationary phase. Overall, a larger number of discriminating metabolites was found in *BPMP-EL-40* than in *BPMP-PU-28* exudates (Figure 6A and Appendix A). This is especially shown in Figure 6B and Appendix A, whose heat map compares the exo-metabolomes of the two bacterial species grown at the stationary phase in buffered medium, i.e., the exo-metabolomes present in the same culture supernatants previously tested for their effects on root development (Figure 3 and Figure 4).

### 2.6. Comparative Proteomics Profiling of BPMP-PU-28 and BPMP-EL-40 Exudates

A bottom-up approach using nanoLC-MS/MS fragmentation was performed to analyze bacterial sub-proteomes present in *BPMP-PU-28* and *BPMP-EL-40* growth supernatants.

The bacteria were grown in buffered growth medium, and their supernatants were collected during the stationary phase, alongside supernatants whose effects on root system development were previously studied (Figure 3 and Figure 4). The collected samples were either digested or not for proteins or peptides, respectively, with trypsin and directly analyzed after a desalting step. The identified proteins/peptides, 124 for *BPMP-PU-28* and 259 for *BPMP-EL-40*, are listed in Appendix A (Data Sheet 1). After removing redundancy, about one-third of the identified peptides correspond to outer membrane proteins or periplasmic proteins, 6% belong to ABC transport systems, 6% to proteases, and 5% to redox and stress proteins. About 11% correspond to ribosomal proteins, and another 11% to hypothetical proteins or proteins with DUF (Domain of Unknown Function) domains. Such a composition, and in particular the presence of ribosomal proteins, provided evidence that endocellular proteins were released into the culture medium, at least in part, presumably as a result of cell death. To circumvent this problem, whole bacterial proteomes were analyzed in the same experimental conditions, resulting in the identification of 1664 proteins in *BPMP-PU-28* and 1718 in *BPMP-EL-40* (Appendix A; Data Sheet 2). A list of proteins found in both the culture medium and the “whole proteome” was then compiled, which identified 41 proteins for *BPMP-PU-28* and 102 proteins for *BPMP-EL-40* (Appendix A, Data Sheet 3), representing, respectively, 48% and 33% of the proteins initially identified in the growth supernatants.

The remaining proteins (present in the growth supernatants and not identified in the whole proteome), 83 for *BPMP-PU-28* and 157 for *BPMP-EL-40* (Appendix A, Data Sheet 4), were considered to be more specifically present in the growth supernatant than inside the bacteria, and thus as potentially constituting (part of) the “exo-proteome” of the bacterial species. An overall comparison between this so-called exo-proteome and the entire bacterial proteome was performed by compiling the semantic terms overrepresented in each protein list using the web site (https://www.nuagesdemots.fr/) (online published since 2003). The results (Appendix A) provided evidence that the proteins present in the exo-proteome list constituted a truly specific sub-proteome, compared to the full bacterial proteome.

The proteins present in the exo-proteome list (Figure 7) could be classified into the same set of eight categories for both bacterial species, with the relative size of these categories being different between the two species. The presence of a surprisingly large “proteases” category, about 14% of the proteins in the *BPMP-PU-28* exoproteome and 6% in the *BPMP-EL-40* one, is worth noting, with the percentage of proteases in the total proteomes of these two bacterial species being about 3.5% and 4%, respectively. The “miscellaneous” category includes proteins that belong to various metabolic pathways (e.g., folding, redox…), or that cannot be easily classified. It comprises about half of the exo-proteome in both bacterial species. Within this category, only two common proteins are present in the two bacterial strains. The composition of this category seems, thus, to be “strain specific”.

## 3. Discussion

### 3.1. Bacterial Strains Isolated from the Rhizosphere of a Wheat Ancestor Can Behave as Efficient PGPR in Modern Wheat Varieties

Different bacterial strains, belonging for example to the genera *Azospirillum* [44,45,46], *Bacillus* [47,48], *Pseudomonas* [49,50], or *Enterobacter* [48], have been identified as having PGPR effects on modern wheat varieties. The impact of a PGPR interaction on plant growth—in terms of biomass production or grain yield, either in the laboratory or in the field—depends on the plant species/genotype and bacterial strain [10,22,51,52,53,54], but also on biotic [55], and abiotic conditions, including soil pH, nutrient availability and fertilization levels [54,55,56,57].

In wheat, for example, field tests of a collection of *Azospirillum* strains (one of the best studied genera of plant growth promoting rhizobacteria) at 297 different experimental locations in Argentina showed a growth-promoting effect in about 70% of the cases, which resulted in an increase in grain yield from 13% to 25% for the most effective strains/interactions [57,58]. In Brazil, comparing nine *Azospirillum* strains has led to similar observations, with increases in grain yields most often in the range of 13–18%, and up to 31%, depending on the season and field location [59]. Likewise, a meta-analysis conducted on 59 available articles to evaluate the extent to which *Azospirillum* strains can contribute to wheat growth, revealed a mean increase of 8.9% in seed yield [10]. In plants grown in controlled conditions, in pots in a greenhouse, larger impacts of inoculation with PGPR have been reported. For instance, wheat inoculation with *Azospirillum brasilense* (strain BNM-10), after soil sterilization, was found to result in about a 40% increase in biomass production and 60% increase in grain yield [55].

Our results indicate that the two bacterial strains that we have isolated can behave as *bona fide* PGPRs. These strains have been shown to stimulate plant growth when the availability of assimilable N in the soil is limiting for growth (when compared with growth in a nitrate-rich control soil), but still sufficient to allow uninoculated plants to complete their cycle from grain to grain. Under these conditions, *BPMP-PU-28* and *BPMP-EL-40* increased total grain mass (harvested per pot) by 90% and 53%, respectively, and the total amount of N in the produced grains (per pot) by about 98% and 128% for *BPMP-PU-28* and *BPMP-EL-40*, respectively (Figure 2). These results therefore indicate that both *BPMP-PU-28* and *BPMP-EL-40* can actually behave as efficient PGPRs in wheat. The increases in yield and total seed nitrogen content, resulting from inoculation with either *BPMP-PU-28* and *BPMP-EL-40*, are, however, far below those resulting from plant watering with nitrate-rich solution (Hoagland solution, 11 mM of NO_3_^−^) (Figure 2F,I). Similar differences in growth promotion were also observed between *Azospirillum lipoferum* (a well-known PGPR) and the nitrate-rich solution (Appendix A). Thus, high levels of fertilization with assimilable N sources were found to be much more efficient than PGPRs in terms of plant nutritional supply and plant growth, in our experimental conditions. These results suggest that diazotrophs are not as effective as high levels of nitrogen fertilizers in providing nitrogen and supporting plant growth. Of course, however, it can be assumed that under most environmental conditions, plant growth promotion by PGPR results from many diverse bacterial activities and not solely from improvement in plant nitrogen nutrition. Besides nitrogen fixation (Figure 1), properties classically proposed to contribute to plant growth and health promotion by PGPRs have been found in the two strains *BPMP-EL-40* and *BPMP-PU-28*. These properties include auxin production in the presence of tryptophan (larger in *BPMP-EL-40*; Appendix A), hydrogen cyanide production (in *BPMP-PU-28*; Appendix A), and the ability to solubilize poorly soluble sources of phosphate and potassium (larger in *BPMP-PU-28*; Appendix A).

There is evidence available proving that the process of crop domestication has affected root microbiome features in a plant genotype-dependent manner [22,60,61]. Millions of years of co-evolution between crop ancestors and their microbiota had favored the development of specific and beneficial interactions, which were then altered by domestication and selection. Especially, plant breeding for high yields in artificialized soil conditions, under high fertilizer inputs, is thought to have impacted plant traits involved in beneficial plant–microbe interactions [61]. Wheat was domesticated in the Fertile Crescent area, which includes Lebanon. Our results show that a bacterial strain, *BPMP-PU-28*, isolated from the rhizosphere of a wheat ancestor spontaneously growing in a refuge area in Lebanon, and another strain *BPMP-EL-40*, isolated from the rhizosphere of the same species but grown far from the Fertile Crescent, in France, can interact with a modern wheat variety and behave as *bona fide* PGPRs. Altogether, these results indicate that the capacity of wheat to establish efficient beneficial interactions with PGPRs has not been profoundly and systematically modified by domestication and breeding.

### 3.2. BPMP-PU-28 and BPMP-EL-40 Exudates Modify Root System Development

Root system development over seven days was affected by supplementing the nutrient solution with *BPMP-PU-28* or *BPMP-EL-40* exudates (Figure 3 and Figure 4). Plant inoculation with living PGPRs has been extensively reported to impact root system development, which is thought to contribute to improving plant hydro-mineral nutrition [2,62,63,64]. In Arabidopsis, inhibition of primary root growth, stimulation of lateral root production, increased length of lateral roots, and strong promotion of root hair elongation appear to be almost systematically induced [7] by very different PGPR strains [19,65,66,67,68,69,70,71] with exceptions since, for instance, a promotion of primary root growth has also been reported [47]. Less information is available on wheat and it mainly concerns bread wheat *T. aestivum*. Contrasting results have been reported in this species depending on the experimental conditions, especially the level of inoculation and possibly the PGPR strain. Indeed, inoculation with different PGPR species, including *A. brasilense* strains, has been reported to result in an increase in the total length of the root system [45,48], or in a strong decrease in root length, associated to thicker roots (larger root diameters), and a strong increase in root hair density and length [46]. The latter developmental responses of the root system, observed when the inoculation level was high, have been found to involve auxin production by the inoculated bacteria [46].

The present results indicate that adding aliquots of *BPMP-PU-28* and *BPMP-EL-40* culture supernatants into the wheat seedlings’ nutrient solution resulted in altered root system development, with distinctive responses for the two strains (Figure 3 and Figure 4). An increase in root system biomass in response to *BPMP-EL-40* exudates was observed, while *BPMP-PU-28* exudates were without significant effect on this parameter. A reduction in total root length and an increase in mean root diameter was observed in response to both culture supernatants, the latter response being, however, significantly larger in the case of *BPMP-EL-40* (Figure 3). Close to root apices, a strong increase in root hair length and density was observed in response to *BPMP-PU-28*, and not to *BPMP-EL-40* culture supernatant (Figure 4). Such root responses to the culture supernatant free from bacteria are reminiscent of the observations reported by [46] using living bacteria. The fact that responses to bacterial culture supernatants can be straightforwardly observed, may provide a way, by biochemical fractioning of the growth media, to identify bacterial metabolites and/or peptides that induce root system developmental changes. It is worth noting that, although both *BPMP-PU-28* and *BPMP-EL-40* can produce and secrete IAA when the growth medium is supplemented with tryptophan (Appendix A), the present metabolomics analyses have not identified any type of auxin (or any other kinds of phytohormone families such as cytokinins or gibberellins) in the culture supernatants (Appendix A). This suggests that compounds other than auxin were responsible for the observed developmental responses of the root system to the bacterial exudates.

### 3.3. The High Complexity of the Exo-Metabolomes and Exo-Proteomes Offer Extensive Communication and Action Possibilities for Bacteria

Metabolomes and proteomes have been obtained on total extracts of PGPR bacteria [72,73,74,75,76] but very few data on exo-metabolomes [76], and none on exo-proteomes, have been reported. Our data show the presence of a large variety of metabolites and peptides in the exudates of *BPMP-PU-28* and *BPMP-EL-40*. The composition of these external metabolomes and proteomes strongly depends on the bacterial species, of course, but also on the characteristics of the medium (pH) and the phase (exponential or stationary) of the culture. A total of 108 metabolites with a score equal to or higher than 6.5 were identified in the metabolomes of *BPMP-PU-28* and *BPMP-EL-40*. The corresponding values for the proteomes of *BPMP-PU-28* and *BPMP-EL-40* are 125 and 214 sequences, respectively, of which 110 and 84 appear to be found more specifically in the culture supernatants than in the total cellular proteomes—thus belonging to what we have operationally defined as the bacterial “exoproteomes”.

The PGPR exo-metabolome analysis reported by [76] concerns two different strains of *Pseudomonas*. For each strain, about 110 metabolites were found in the growth supernatant. This differed from our own minimal growth medium since it was supplemented with 15 different amino acids and contained fructose as a carbon source, which may hamper the identification of metabolites with a molecular weight below about 200 Da (molecular weight of fructose: 180 Da). The number of compounds (about 110) found in the growth media for each strain under these conditions is, however, quantitatively consistent with our own analyses. The lists of identified metabolites were not provided, but the authors mentioned the presence of N-acyl homoserine lactone (AHLs). Of particular note is that the AHL N-tetradecenoyl-L-homoserine lactone (TDHL) has also been found in *BPMP-PU-28* and *BPMP-EL-40* exudates (Figure 5, Appendix A; compound 341). AHLs are amphiphilic molecules with a hydrophilic homoserine lactone ring and a hydrophobic side acyl chain [77]. The length of this acyl chain can vary from 4 to 18 carbon atoms and generates specificity between bacteria [43]. AHLs play a role in bacterial quorum sensing (QS) and in bacterial communication networks. They have also been shown to have inter-kingdom signaling properties. They have positive effects on plant growth [78,79], and could be recognized by plant receptors and lead to modifications of plant gene expression [80,81].

It should also be noted that while no amino acids were detected in the bacterial exudates under our experimental conditions, different cyclopeptides were present, namely the cyclodipeptide (CDP) cyclo(L-Pro-4-OH-L-Leu) (cycloHPL), a nucleoside peptide named Nikkomycin Wx (composed of L-tyrosine and 5-amino-5-deoxy-D-allo-furanuronic acid N-glycosidally bound to 4-formyl-4-imidazolin-2-one) [82], and the cyclic tetrapeptide Cyclo-(Tyr-Ala-Leu-Ser) (or brevibactin A) (Figure 5, Appendix A). Together with AHLs, cyclic peptides have been shown to play a role in quorum sensing [83,84]. The higher abundance of AHLs and cyclic peptides in the exudates collected in stationary phase (Appendix A) is consistent with the fact that part of the bacterial population was then engaged in a biofilm lifestyle, as also indicated by the presence of a veil on the walls of the Erlenmeyer flasks [85]. In addition to playing a role in quorum sensing, cyclic peptides can act as mimetics of phytohormones [79,86,87]. The CDP cyclo (L-Pro-4-OH-L-Leu) found in *BPMP-PU-28* and *BPMP-EL-40* growth supernatants is close to the CDP cyclo (L-Leu-L-Pro) identified in *Bacillus gaemokensis* and is shown to upregulate salicylic acid, ethylene and jasmonic acid signaling [88]. It is also worth noting that the CDPs cyclo (L-Pro-L-Val), cyclo (L-Pro-L-Phe) and cyclo (L-Pro-L-Tyr), which are produced by different *Pseudomonas* strains (*P. aeruginosa*, *P. putida* and *P. fluorescens*), have been reported to have auxin-like activity in Arabidopsis, and to modulate auxin-responsive gene expression in roots, suggesting a role of bacterial cyclodipeptides as phytostimulants [3,89].

In addition to AHLs and cyclopeptides, many of the metabolites identified in *BPMP-PU-28* and *BPMP-EL-40* exudates may play a role in plant growth promotion, for example by behaving as antibiotics (about 18% of the identified metabolites can be expected to have antibiotic effects), or by improving nutrient ion acquisition, such as coproporphyrin III which is one of the compounds identified in both strains (compound 59 in Appendix A).

With respect to the proteomics data, the exo-proteomes identified in the present study highlight and confirm the routes and structural components of bacterial exudation. Indeed, many outer membrane proteins (OMPs) such as porins, Type I, IV, VI secretion system proteins, flagellar and fimbrial proteins, periplasmic proteins and lipoproteins have been identified, but no inner membrane proteins. More than a third of the bacterial proteome is likely to be extra-cytoplasmic [90], located in membranes in the periplasmic space, or secreted. Different models have been proposed for bacterial exudation especially through outer-membrane vesiculation [91,92,93,94]. Production of outer-membrane vesicles (OMVs) underlies evolutionarily conserved mechanisms important in cell communication. OMVs can carry metabolites, enzymes, peptides and nucleic acids. They can deliver high concentrations of active compounds, allow protection of transported molecules [95,96], and have been shown to play a role in horizontal gene transfer, defense, virulence and intra- and inter-species communication [97,98]. It is interesting to note that the same categories of outer membrane components are found in both *BPMP-PU-28* and *BPMP-EL-40* exudates (including proteases), revealing a common pathway for exudation. Besides these common OMPs, the remaining exo-proteome components, which are likely to be mainly transported in the lumen of OMVs, are species specific. A striking result is the overrepresentation of the category “peptidase”, which comprises 14% and 6% of the proteins in the *BPMP-PU-28* and *BPMP-EL-40* exoproteomes, respectively. It is also interesting to note that, among the semantic terms describing the *BPMP-PU-28* and *BPMP-EL-40* exoproteomes (Appendix A), the presence of “Substrate Binding” indicates a type of membrane transport activity that is likely to contribute to solute exchange between the bacteria and the host plant roots.

### 3.4. Conclusions

Thus, our data provide evidence that PGPRs can constitutively produce very rich and complex exo-metabolomes and exo-proteomes, the composition of which is significantly dependent on the external environment and the bacterial lifestyle (planktonic phase or biofilm). The richness and complexity of these metabolomes and proteomes support the hypothesis that, in each PGPR strain, numerous and complex mechanisms can simultaneously contribute to plant growth promotion. Figure 8 provides a synoptic view of the diverse pathways and mechanisms that the exuded compounds could trigger, giving rise to beneficial interactions with the plant.

Exudates are likely to be protected in outer-membrane vesicles, which allow long distance communication. Proteases and exo-metabolites, such as antibiotics or chitin synthase inhibitors, are potentially involved in defense against competitors, plant pathogens and pests. Other exo-metabolites such as riboflavins, coproporphyrin and dethiobiotin are likely to be involved in plant nutrition, while specific cyclopeptides and N-acyl-homoserines lactones can potentially mimic phytohormones and affect root and root hair development.

All the exo-metabolites and proteins whose names are present in this figure can be retrieved with their number in Appendix A, respectively.

This diversity is also likely to broaden the spectra of plant genotypes, cultivars, accessions and species with which beneficial interactions can be developed. It might contribute to the fact that PGPR strains isolated from the rhizosphere of a wheat ancestor growing in its refuge area in Lebanon, or far from this region in a west Europe soil, can display beneficial effects upon interactions with a modern wheat elite cultivar. It would be worth further deciphering and evaluating these mechanisms and pathways, by investigating the effects of host plant root exudates on the bacterial exoproteomes and metabolomes. Furthermore, the fact that collected bacterial exudates can, in vitro, affect the development of the root system, is likely to help, by biochemical fractionation, identify the mechanisms involved in root–PGPR communication and the resulting benefits for the plant.

## 4. Materials and Methods

### 4.1. Soil Sampling

Wild emmer (*T. t. dicoccoides*, accession Ttd-NC-2019), a wheat ancestor, was grown in two soils—one from a non-cultivated refuge area of the Fertile Crescent in Lebanon (latitude: 33°51′35.6′′ N: longitude: 36°07′09.6′′ E; altitude: 1220 m) near the town of Nabi Chit, and the other one from France (latitude: 43°37’11.0” N; longitude: 3°58’45.2” E; altitude: 12 m) near the town of Mauguio. The two soils were sampled in 2019 and had similar physical and chemical properties (Appendix A) based on analyses carried out as described in [99].

### 4.2. Diazotrophic Bacteria Selection and Characterization

Bacteria were isolated from the entire rhizosphere (grinded soil + roots) of *T. t. dicoccoides* and grown for 6 weeks in both soils, in a growth chamber with a 16-h diurnal photoperiod, a day/night temperature of 22/20 °C, a light intensity of 150 μE, and a relative air humidity of 70%. Diazotrophic bacteria were isolated using semi-solid selective Nitrogen Free Bromothymol Blue (NFb) medium according to [100]. Briefly, rhizospheric samples (ca. 10 mg of soil) were surface-inoculated in semi-solid NFb tubes. About three days later, sub-surface veils were collected and inoculated again in new semi-solid NFb tubes to confirm bacterial growth in this N-free medium. The last sub-surface veils were then serially diluted and inoculated by spread-plating onto solid NFb medium, supplemented with 40 mg/L yeast extract. Diazotrophic strains were then individually identified amongst the colonies, as those able to form sub-surface veils in semi-solid NFb tubes. After extraction of genomic DNA (Neo-Biotech Quick Bacteria Genomic DNA Extraction Kit), bacterial isolates were first distinguished based on their BOX- profiles [101] and then further identified by 16S rRNA gene sequencing [102]. Assessment of bacterial ability to solubilize and use (i) poorly soluble sources of P (tri-calcium phosphate and phytate) was performed using Pikovskaya medium [103], and (ii) a poorly soluble source of K^+^ (feldspar, potassium aluminosilicate, potash feldspar, Bath Potters’ Supplies, Somerset, UK) was performed using Alexandrov medium, according to [104]. The solubilization index (SI; (colony diameter + halo zone diameter) divided by colony diameter; [99,100]) was determined 5 days after the bacterial suspension was dropped on the agar medium (incubation at 28 °C). Indole-3-acetic acid production was determined according to [105] and [106]. Hydrogen cyanide production was tested according to [105] and [107].

Bacterial capacity to fix atmospheric nitrogen was assessed by injecting 2 mL of ^15^N_2_ into a 9 mL tube (Vacuette^®^ tube, 455001-Greiner Bio-One), through the septum of the tube, containing 3 mL of liquid NFb medium. After 5 days of culture at 37°C, the *δ*^15^N (^15^N/^14^N) was assayed in freeze-dried bacterial pellets obtained after centrifugation (10 min at 3220 rcf at room temperature) using isotopic mass spectrometry (Elemental Analyzer Vario-PYROcube coupled to an IsoPrime Precision mass spectrometer, Elementar, Langenselbold, Germany).

From an initial collection of 32 diazotrophic bacterial strains (16 strains isolated from the Lebanese soil and another 16 strains from the French soil), two species displaying distinctive characteristics according to the tests described above, one from the Lebanese soil (LS) and the other from the French soil (FS), were selected for the present study. After genome sequencing, they were matched to *Pseudomonas urmiensis* (LS) and *Enterobacter ludwigii* (FS) and were named *BPMP-PU-28* and *BPMP-EL-40*, respectively, in this manuscript.

Bacterial whole-genome sequencing was performed at Beijing Genomics Institute (BGI, Hong Kong, China) using dnbseq sequencing technology [108]. Contigs assembly and gene and protein annotation were carried out at https://galaxy.migale.inrae.fr/ (2018) and the genomic maps were built using the BLAST Ring Image Generator (BRIG software) [109].

### 4.3. PGPR Effect on Growth and Development of an Elite Wheat Cultivar

Seeds of an elite durum wheat (cv. Anvergur) were surface-sterilized (by soaking for 20 min in 4 % calcium hypochlorite) and rinsed 4 times for 5 min with sterile distilled water under a laminar flow hood. They were transferred onto sterile water-humidified filter paper in a Petri dish and kept in the dark at 26 °C for 2–3 days until they had a visible hypocotyl and roots 3–4 cm long. Seedlings were inoculated by immersion for 1 h in a bacterial suspension (10^7^ CFU.mL^−1^ in water; OD_600_: 0.1). They were then immersed in sterile water and thereafter transferred onto a sterilized substrate (peat/sand/vermiculite mixture, 1:1:1 *v*:*v*:*v*, in 3 L pots; 4 seedlings per pot; 6 pots per condition, placed in a tray moved randomly every week in the greenhouse) in the greenhouse (16 h diurnal photoperiod, a day/night temperature of 21/17 °C). Uninoculated seedlings treated in a similar way were used as controls. The pots were watered with a Hoagland solution (1 mM NH_4_NO_3_, 5 mM KNO_3,_ 2 mM Ca(NO_3_)_2,_ 2 mM MgSO_4_, 1 mM KH_2_PO_4_, 0.1 mM NaFe(III) EDTA, 12.5 µM H_3_BO_3_, 2 µM MnCl_2_, 3 µM ZnSO_4_, 0.5 µM CuSO_4_, 0.1 µM Na_2_MoO_4_, 0.1µM NiSO_4_ and 25 µM KCl) as used by [35], or with modified Hoagland solutions containing a low concentration of nitrate, either 100 µM or 250 µM KNO_3_ as a unique nitrogen source (no addition of NH_4_NO_3_ in the nutrient solutions, 2 mM Ca(NO_3_)_2_ being replaced with 2 mM CaCl_2_, and the reduction of the concentration of KNO_3_ being compensated for by addition of KCl).

The plants were phenotyped using the PhenoArch platform [110] hosted at M3P, Montpellier Plant Phenotyping Platforms (https://www6.montpellier.inrae.fr/lepse/Plateformes-de-phenotypage/Montpellier-Plant-Phenotyping-Platforms-M3P) (accessed on 11 June 2020). RGB images were taken for each plant from 13 views (12 side views with 30° rotational difference and one top view). Briefly, plant pixels were segmented from the background using a combination of thresholding and random forest algorithms, as described by [111], and available at (https://github.com/openalea/phenomenal) (accessed on 16 November 2020). The whole plant leaf area and shoot fresh weight were estimated using calibration curves built with multiple linear regression models, based on processed images against ground truth measurements of leaf area and fresh biomass.

The number of spikelets per rachis node on the main shoot was determined after 11 weeks of growth. Thousand-seed mass and the total seed mass per pot were determined at the end of the cycle. Seeds were harvested, shelled and then ground with a mortar. The seed nitrogen and carbon contents of the samples were determined using isotopic mass spectrometry (Elemental Analyzer Vario-PYROcube coupled to an IsoPrime Precision mass spectrometer, Elementar, UK).

### 4.4. Bacterial Culture Media for Metabolomic and Proteomic Profiling

Preliminary experiments were performed to develop a minimal medium containing a carbon source with a molecular mass below 100 Da, for the purpose of investigating the exo-metabolomes of PGPR strains. Lactic acid (molecular weight: 90 Da) was chosen because it was found to allow most of the tested PGPRs from our collection, including *BPMP-PU-28* and *BPMP-EL-40*, to grow at a sufficiently rapid rate. Two types of bacterial culture media were used, buffered and unbuffered. The basal medium was a Hoagland solution (composition detailed above) complemented with 2% lactic acid (approximately 220 mM) as the sole carbon source. It was either supplemented with 150 mM phosphate buffer (KH_2_PO_4_/K_2_HPO_4_, pH 6.8) or not supplemented with buffer (pH adjusted in both cases to 6.8 with NaOH).

### 4.5. Bacterial Growth Curve in Minimal Media for Metabolomic and Proteomic Analyses

Half a loop of a bacterial colony, grown on LB medium in a Petri dish, was transferred into an Eppendorf tube containing 300 μL of either buffered or unbuffered minimal growth medium. A volume of 30 μL of the bacterial suspension thereby obtained was transferred into a pre-culture tube containing 3 mL of the corresponding minimal medium. Pre-culture was performed at 37 °C with gentle shaking at 200 rpm. After approximately 24 h, 300 μL of the bacterial pre-culture (approximatively 10^9^ colony-forming units, CFU) was transferred into a 200 mL Erlenmeyer flask containing 50 mL of the corresponding minimal growth medium. The inoculated medium in the Erlenmeyer was then incubated for approximately 60 h at 37 °C and 200 rpm. The bacterial growth curve was generated by measuring the optical density of the bacterial suspension at 600 nm every two hours (Spectrophotometer, SmartSpec™ 3000–BIO-RAD, Hercules, CA, USA).

### 4.6. Metabolomics and Proteomics Profiling of Bacterial Exudates

Samples from the above-described bacterial cultures were taken during both the exponential and stationary phases of the growth, with respect to the growth curves, after 24 h and 48 h of culture for *BPMP-PU-28*, and 16 h and 40 h for *BPMP-EL-40*, respectively. After two successive centrifugations (3220 rcf, 15 min at 4 °C), supernatants were filtered (0.22 μm filter) and were either directly desalted using a C18 Sep Pak column (WAT036820-SPE Cartridge, SEP-PAK ^®^ TC18 Cart 1cc, Waters^TM^, Milford, MA, USA) according to [112] for metabolomics analysis, or first reduced with 10 mM DTT (1.4-dithiothreitol), alkylated with 50 mM iodoacetamide, and then desalted using the same type of C18 Sep Pak column for proteomics analysis. Column eluents were lyophilized and stored at −20 °C before metabolomics and proteomics analyses.

For metabolomics, ultra-high-performance liquid chromatography−high-resolution MS (UHPLC−HRMS) analyses were performed on a Q Exactive Plus quadrupole mass spectrometer. Data processing and annotation were performed by MS-CleanR workflow [37]. See Appendix A.

For proteomics, the samples were either digested or not with trypsin. They were analyzed online using nanoHPLC (NCS3000, Thermo Fisher, Waltham, MA, USA), with a gradient of 140 min, coupled to a mass spectrometer having the nano electrospray source Q-Exactive plus (Thermo Fisher Scientific). Database-dependent search algorithms and de novo sequencing were performed using the PEAKS X plus software (Bioinformatics Solutions Inc., Waterloo, ON, Canada) [113]. The database of peptides, deduced from the 6 reading frames generated from each bacterium genome, was used for peptide and protein identifications. Cysteine carbamidomethylation (+57.02 Da) was set as a static modification, and methionine oxidation (+15.99 Da) and deamidation (+0.98 Da), as a variable modification. The parent mass error tolerance was set to 10 ppm, and the fragment mass error tolerance was set to 0.05 Da. For trypsin digestion, maximum missed cleavages were set at 3 and for non-digestion, no enzyme and unspecific digestion were used. The false discovery rate (FDR) threshold was set to 1%. Peptides identified by de novo sequencing were blasted (https://blast.ncbi.nlm.nih.gov/Blast.cgi?PAGE=Proteins) (2009). Raw data were deposited on PRIDE-Proteomics Identification Database (PXD034914).

### 4.7. Effects of Bacterial Growth Supernatants on Wheat Development in Hydroponics Condition

Young, sterilized seedlings of the elite durum wheat (cv. Anvergur) were obtained according to the protocol described above. After excision of the seeds, they were transferred onto 40 mL of sterile Hoagland solution supplemented with sterile (0.22 µm-filtered) growth supernatants taken during the stationary phase of the bacterial growth (final dilution: 5%). The final pH of the solution was adjusted to 6 with HCl. For mock treatment, seedlings were grown in the same hydroponic conditions using a Hoagland solution supplemented with 5% of a solution containing 0.8% of lactic acid, the Hoagland salts listed above, and 150 mM phosphate buffer. The growth solution was contained in 50 mL glass tubes. One seedling was introduced per tube, and a piece of sterile cotton was arranged around the coleoptile and pressed into the tube (without contact with the hydroponics solution), in order to preserve the sterility of the solution while allowing gas exchanges with the atmosphere. The entire procedure was carried out under a laminar flow hood. The tubes were wrapped in aluminum foil (to protect the growing roots against light) and placed in the culture chamber for 7 days. The sterility of the hydroponics solution at the end of the 7-day growth was tested using LB agar plates, and detection of microorganisms on the plate after 48 h of incubation at 37 °C led to discarding the corresponding plant. Plant root systems were scanned (Epson Perfection V850 Pro Scanner, Epson, Nagano, Japan). The resulting images were analyzed to measure root system parameters using the WinRHIZO^TM^ software (V.2009 Pro, Regent Instruments, Montreal, QC, Canada), as indicated (http://regent.qc.ca/assets/winrhizo_software.html (released in 1996) for “Analysis of washed root systems”).

### 4.8. Statistical Analysis

For biochemical and physiological tests, data are mean values of independent experimental repetitions. Depending on the experiments, differences among treatments were analyzed by unpaired t-test or one-way ANOVA, followed by Tukey’s multiple range, using GraphPad Prism 8 version 8.3.0 (San Diego, CA, USA) and taking *p* ≤ 0.0001, *p* ≤ 0.001, *p* ≤ 0.01 or *p* ≤ 0.05, based on unpaired experimental design, as significant.

For metabolomics data, statistical analyses were performed with metaboanalyst 5.0 web interface [114]. All data were normalized to total ion chromatogram (TIC) and were UV (unit variance) scaled before multivariate analysis.

## Figures and Tables

**Figure 1 ijms-23-15248-f001:**
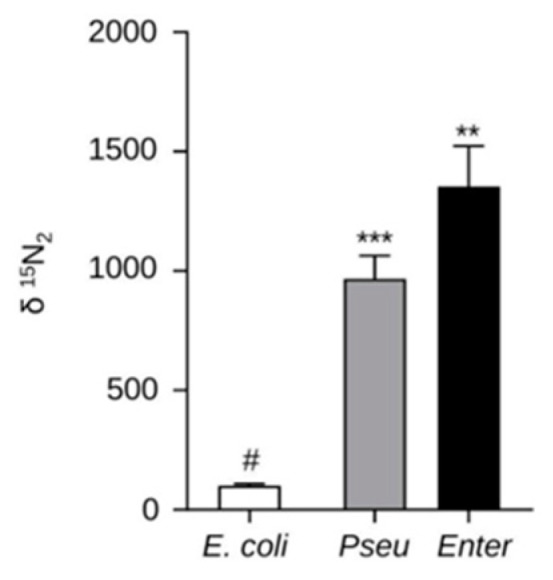
Nitrogen fixation ability of the selected diazotrophic bacterial strains. ^15^N_2_ incorporation was used to assess the N_2_ fixation capacity of *BPMP-PU-28* (Pseu: *Pseudomonas urmiensis*) and *BPMP-EL-40* (Enter: *Enterobacter ludwigii*) strains. *E. coli* (5-alpha F’Iq *E. coli*, New England Biolabs) was used as the negative control. The gas phase (6 mL) above the bacterial suspension (3 mL) contained about 16% ^15^N_2_, 65% ^14^N_2_ and 18% O_2_. Cells were centrifuged after 5 days of incubation (at 37 °C and 200 rpm), and the δ ^15^N_2_ in the bacterial pellets was determined. Means ± SE (n = 3). ** and *** above the bars indicate that the difference with the negative control (*E. coli*; #), is statistically significant (Student’s *t*-Test, *p* ≤ 0.01 and 0.001, respectively).

**Figure 2 ijms-23-15248-f002:**
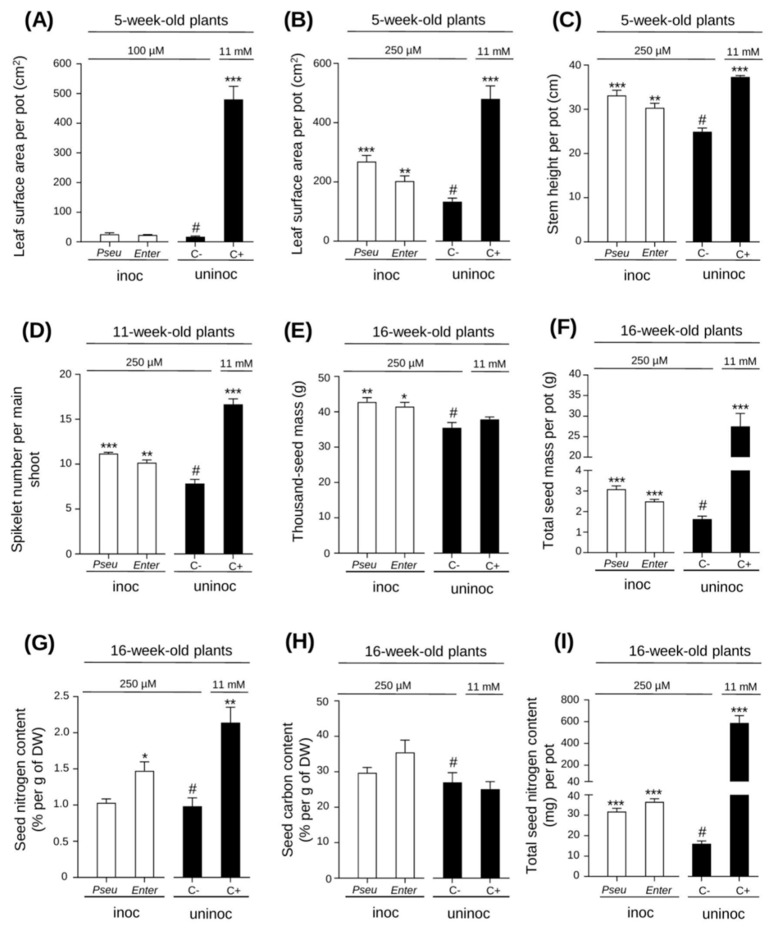
Effect of inoculation with *BPMP-PU-28* (*Pseudomonas urmiensis*) and *BPMP-EL-40* (*Enterobacter ludwigii*) on plant development under low assimilable nitrogen availability. Elite durum wheat cultivar Anvergur plants were grown in a greenhouse in pots on an artificial solid substrate (6 pots per condition, 4 plants per pot). They were inoculated with one of the two selected strains, *BPMP-PU-28* (Pseu: *Pseudomonas urmiensis*), *BPMP-EL-40* (Enter: *Enterobacter ludwigii*), or not inoculated (Uninoc). Nutrients necessary for plant growth were brought with 3 different watering solutions, all derived from Hoagland medium but containing either 100 or 250 µM of assimilable nitrogen (provided as NO_3_) for inoculated plants (white bars) or non-inoculated “negative” control plants (C−, black bars), or containing 11 mM of assimilable nitrogen (10 mM NO_3_ and 1 mM NH^4+^) for uninoculated “positive” control plants (C+, black bars). Leaf surface area and leaf sheath height were measured by the Phenoarch phenotyping platform. (**A**,**B**) Leaf surface area of 5-week-old plants. The plants watered with 100 µM NO_3_ (**A**) were chlorotic and died during the following week. The plants watered with 250 µM NO_3_ (**B**) developed and produced seeds. (**C**) Leaf sheath height (5-week-old plants). (**D**) Spikelet number per main shoot (11-week-old plants). (**E**) Thousand-seed mass (16-week-old plants). (**F**) Total seed mass per pot. (**G**) Seed nitrogen content (% per g of DW). (**H**) Seed carbon content (% per g of DW). (**I**) Total seed nitrogen content (mg) per pot. (**E**–**I**): 16-week-old plants. Means ± SE (n = 6 pots). *, **, and *** above the bars indicate that the difference with the uninoculated condition under nitrogen limitation (#, negative control: C−) is statistically significant (Student’s *t*-Test, *p* ≤ 0.05, 0.01, 0.001, respectively).

**Figure 3 ijms-23-15248-f003:**
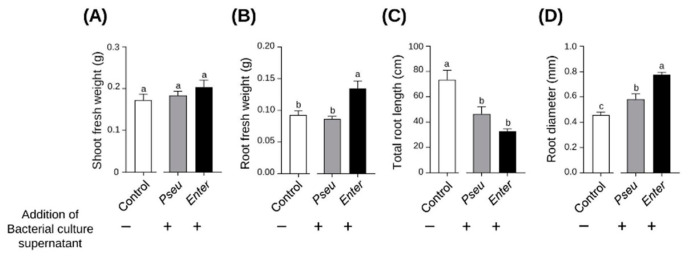
Effects of *BPMP-PU-28* (*Pseudomonas urmiensis*) and *BPMP-EL-40* (*Enterobacter ludwigii*) culture supernatants on root system development. Durum wheat cv. Anvergur was grown hydroponically in Hoagland solution under axenic conditions. The solution (initial volume: 38 mL) was supplemented with 2 mL of bacterial culture supernatant prepared (filtered to 0.22 µm, thus without bacteria) from a culture (buffered growth medium collected during the stationary phase; see S arrows in Appendix A) of *BPMP-PU-28* (Pseu: *Pseudomonas urmiensis*) (gray bars) and *BPMP-EL-40* (Enter: *Enterobacter ludwigii*) (black bars), or was supplemented with 2 mL of a mock medium (not used for bacterial growth: control; white bars). (**A**) Shoot fresh weight, (**B**) root fresh weight, (**C**) total root length and (**D**) root diameter after 7 days of growth. Total root length and diameter were determined using WinRHIZO. Means ± SE (n = 5). Different letters indicate statistically significant differences (one-way ANOVA test, *p* ≤ 0.05).

**Figure 4 ijms-23-15248-f004:**
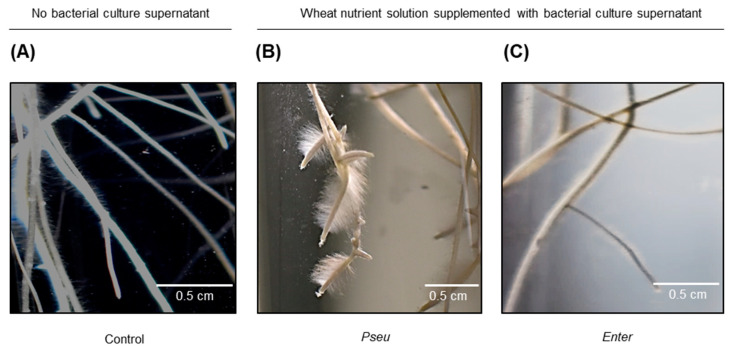
Root hair development induced by *BPMP-PU-28* (*Pseudomonas urmiensis*) culture supernatant in lateral roots of durum wheat seedlings cv. Anvergur grown in hydroponics. Durum wheat cv. Anvergur was grown hydroponically under axenic conditions in Hoagland solution, supplemented with bacteria culture supernatant, or mock medium as described in the legend to Figure 3. Representative images of root systems of plants grown for 7 days in the control solution (**A**), or in the solution supplemented with 5% of *BPMP-PU-28* (Pseu: *Pseudomonas urmiensis*) culture supernatant (**B**) or supplemented with 5% of *BPMP-EL-40* (Enter: *Enterobacter ludwigii*) culture supernatant (**C**).

**Figure 5 ijms-23-15248-f005:**
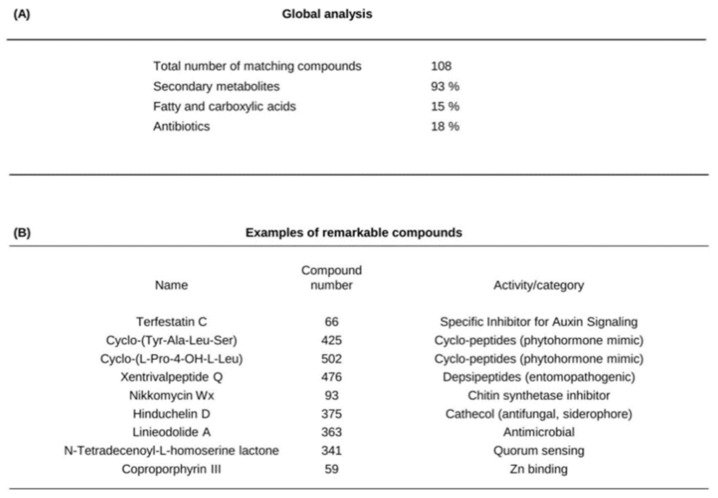
Summary of the metabolomics analysis. (**A**) Global analysis: Total number of matching compounds (from Appendix A) and relative abundancy (% with respect to the total number of compounds) of selected categories of compounds. (**B**) List of selected remarkable compounds and putative activities (from Appendix A).

**Figure 6 ijms-23-15248-f006:**
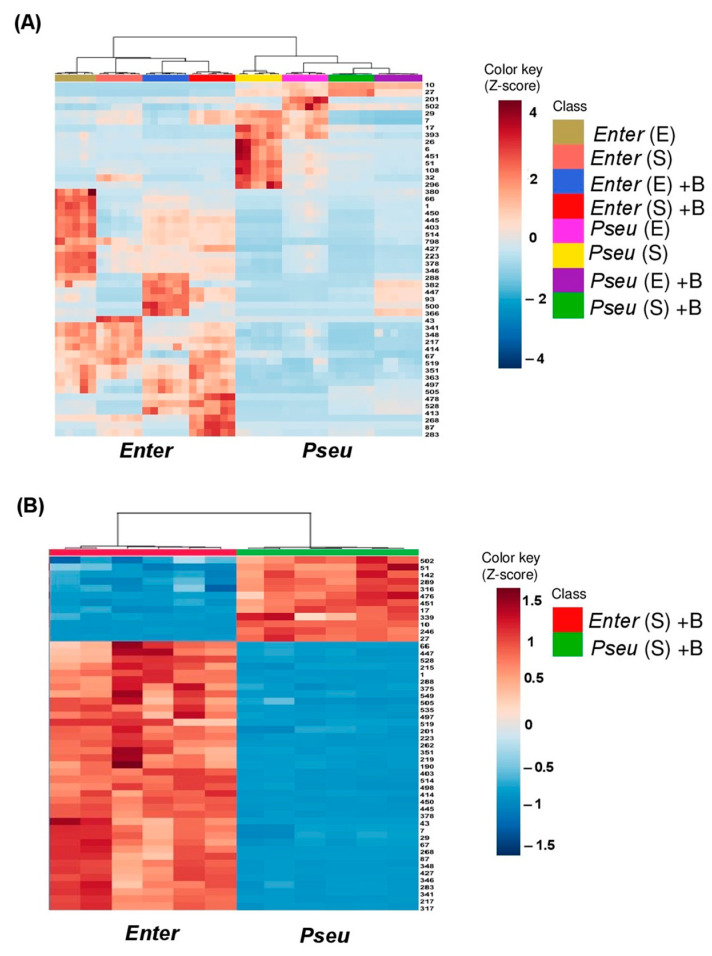
Heat map describing the distribution of the top 50 more discriminant metabolites, between *BPMP-PU-28* (*Pseudomonas urmiensis*) and *BPMP-EL-40* (*Enterobacter ludwigii*), according to culture conditions and growth stages. Pseu (*Pseudomonas urmiensis*) and Enter (*Enterobacter ludwigii*) were grown in minimal Hoagland medium supplemented with lactate (2%) (see Appendix A) and buffered (150 mM KH_2_PO_4_/K_2_HPO_4_, pH 6.7) (B) or unbuffered (U). Aliquots of culture supernatant were collected during the exponential and stationary phases (E and S, respectively; see arrows in Appendix A) and analyzed by LC-MS/MS. The heat map was generated by MetaboAnalyst 5.0 28 (https://www.metaboanalyst.ca/MetaboAnalyst/ModuleView.xhtml) accessed on 15 January 2021). (**A**) Top 50 more discriminant metabolites between *BPMP-PU-28* and *BPMP-EL-40*. Numbers on the right refer to the metabolites, which are listed in Appendix A. (**B**) Discriminant metabolites identified in the bacterial growth media (buffered and collected during the stationary phase) are shown to affect root system and root hair development in Figure 3 and Figure 4. Numbers on the right: see Appendix A.

**Figure 7 ijms-23-15248-f007:**
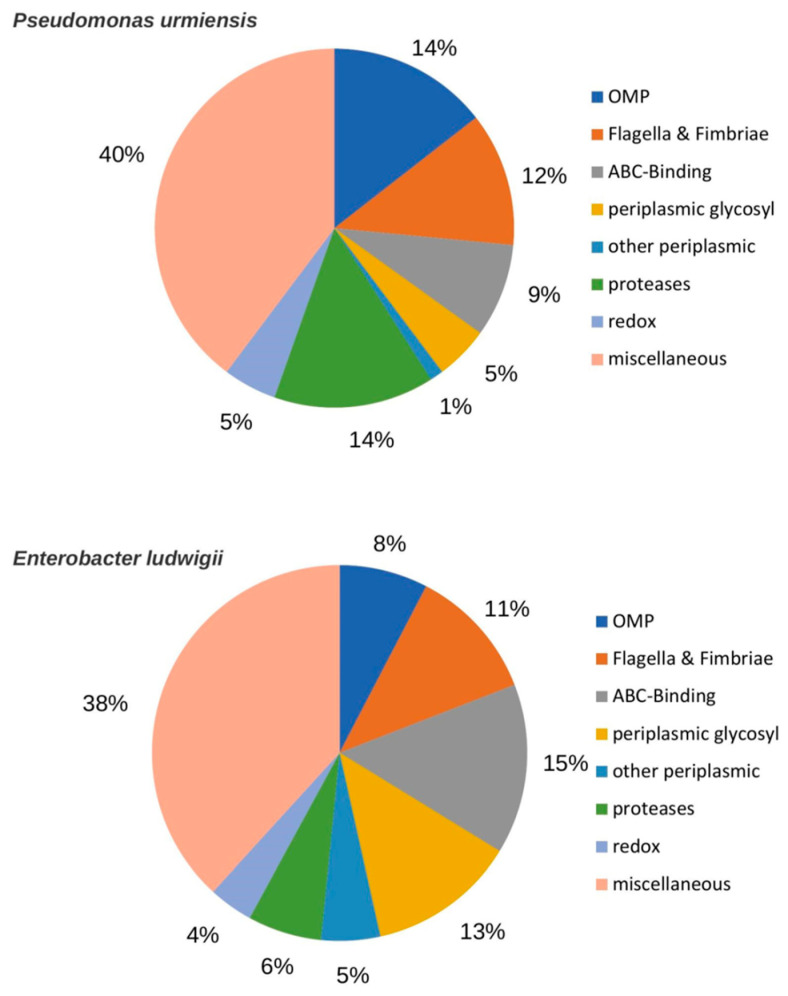
Comparative proteomics profiling of *BPMP-PU-28* (*Pseudomonas urmiensis*) and *BPMP-EL-40* (*Enterobacter ludwigii*) culture supernatant. *Pseudomonas urmiensis* and *Enterobacter ludwigii* were grown in minimal Hoagland medium supplemented with lactate (2%) and buffered (150 mM KH_2_PO_4_/K_2_HPO_4_, pH 6.7). Aliquots of culture supernatant were collected during the stationary phase. Proteins and peptides identified in the growth media and not found in the global proteome of the corresponding bacteria were sorted into 9 classes based on manual annotation (same classes and classification criteria for the two bacterial strains). OMP: outer membrane proteins. Flagella & Fimbriae: proteins belonging to flagella, and long filamentous polymeric protein structures present at the surface of bacterial cells and implicated in adhesion of the bacteria to surfaces, or to other bacteria. ABC substrate: substrate binding proteins of ABC transporters. Periplasmic glycosyl: periplasmic protein implicated in cell wall glycosylation. Other periplasmic: proteins located in the periplasmic space but not involved in glycosylation. Proteases: proteins involved in degradation or maturation of peptides and polypeptides. Redox: proteins involved in oxido-reduction processes. Miscellaneous: proteins with various enzymatic activities and likely to be located neither in the outer membrane, nor in the periplasmic space.

**Figure 8 ijms-23-15248-f008:**
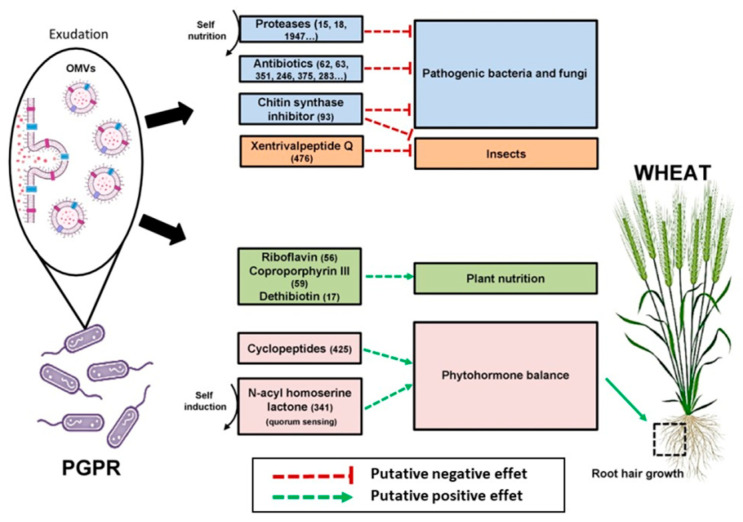
Exo-metabolites and exo-proteins constitutively exudated by *BPMP-PU-28* (*Pseudomonas urmiensis*) and *BPMP-EL-40* (*Enterobacter ludwigii*); diverse and complex pathways are likely to give rise to beneficial interactions with the plant.

## Data Availability

Raw proteomics data and Scaffold file containing all peptide information and spectra are available from the PRIDE repository with accession no. PXD034914. Raw metabolomics data are available from the Zenodo repository (DOI: 10.5281/zenodo.6799532).

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
