# Peer review of "Wild Wheat Rhizosphere-Associated Plant Growth-Promoting Bacteria Exudates: Effect on Root Development in Modern Wheat and Composition"

_ijms, 2022, doi:10.3390/ijms232315248_

Round 1
Reviewer 1 Report
Reviewer’s Comments
The submitted manuscript to IJMS evaluates the effect of plant growth-promoting bacterial exudates on root development and composition in modern wheat. Even though, the topic of manuscript is interesting and under the scope of this journal, however, following are my major concerns before acceptance:
Writing is the main concern for me!!!
For instance: the very first sentence of the abstract is too long and even it is grammatically wrong.
The abstract is lacking the results details. The authors made general statements about M*M rather than their potential results, conclusions and the implications of their study.
The authors should explain the specific the pathway which was affected by the PGPR after investigating several metabolites.
It is suggested to provide the manuscript with line numbering to ease up the review process.
In the second paragraph, the authors again explain the PGPR abbreviation which was already explained in the abstract. This problem should be resolved throughout the manuscript with other abbreviations, too.
Figure 1 and 2: Please provide the letters of significance.
The color selection is very confusing for all the figures.
The authors could choose the following color scheme:
Figure 1: White, grey black
Figure 2: Two white columns (inoc) and two black columns (uninoc)
Figure 3: White, grey and balck
The quality of the figures and tables is very low especially the figure 5.
Figure 5A&B: The quality is very low: even I could not see the names on y-axis. Please provide the high quality figure. For high quality figures, the authors can use meV and tbtools software.
It would also be better to provide the supplementary table for the letters of significance for figure 5A&B so that the readers can see the significant difference among treatments.
Discussion: Please do not use the too much shorter paragraphs.
Conclusion: Avoid using the references, general statements, focus on the main results and their implications.
The most important point: the authors should explain the specific pathway in this study. It is suggested to make the schematic representation of that pathway and the overall study. In addition, this pathway should be mentioned in abstract and conclusion, too.
Author Response
REVIEWER 1
Comment 1.1. Writing is the main concern for me!!! For instance: the very first sentence of the abstract is too long and even it is grammatically wrong.
Answer and changes in the manuscript.
The whole manuscript has been carefully checked and corrected. The first sentence of the manuscript has been shortened, as well as other too long sentences.
Comment 1.2.
The abstract is lacking the results details. The authors made general statements about M*M rather than their potential results, conclusions and the implications of their study.
Answer and changes in the manuscript.
The abstract has been partly re-written to take into account this comment, as well as the comments 2.1 of Reviewer 2.
An important part of the work reported in the present manuscript deals with methodological developments, namely the bacterial growth medium and procedures that have allowed to collect and analyze the exuded metabolites and polypeptides/proteins. We believe that this methodological information will be of interest for many other research groups, and that this will contribute to the citation score of the paper. The corresponding information is therefore of course provided in details in the manuscript. In the abstract, this is summarized in a single sentence "A bacterial growth medium was developed to investigate the effects of bacterial exudates on root development in the elite cultivar and to analyze the exo-metabolomes and exo-proteomes". Based on word or character count, we believe that 89% of the re-written abstract are dedicated to the "results, conclusions and implications" of the study.
Comment 1.3.
The authors should explain the specific the pathway which was affected by the PGPR after investigating several metabolites.
Answer and changes in the manuscript.
A new Figure, namely Figure 8, has been added to the revised manuscript. It provides a synoptic view of the diverse pathways and mechanisms that the exuded compounds could trigger to give rise to beneficial interactions with the plant. This Figure, and the corresponding implications, are briefly discussed in the new Conclusion section (§ 3.4) of the DISCUSSION section. See also below our answer to the comment 1.10.
Comment 1.4.
It is suggested to provide the manuscript with line numbering to ease up the review process.
Answer and changes in the manuscript.
- Done
Comment 1.5.
In the second paragraph, the authors again explain the PGPR abbreviation which was already explained in the abstract. This problem should be resolved throughout the manuscript with other abbreviations, too.
Answer and changes in the manuscript.
- Thanks. Done.
Comment 1.6.
Figure 1 and 2: Please provide the letters of significance.
The color selection is very confusing for all the figures.
The authors could choose the following color scheme:
Figure 1: White, grey black
Figure 2: Two white columns (inoc) and two black columns (uninoc)
Figure 3: White, grey and black
Answer and changes in the manuscript.
- Thanks. Done.
Comment 1.7.
The quality of the figures and tables is very low especially the figure 6.
Figure 6A&B: The quality is very low: even I could not see the names on y-axis. Please provide the high quality figure. For high quality figures, the authors can use meV and tbtools software.
It would also be better to provide the supplementary table for the letters of significance for figure 6A&B so that the readers can see the significant difference among treatments.
Answer and changes in the manuscript.
In Figure 6A&B, all the numbers on the y-axis were manually replaced, with a font as large as possible, allowing an easier reading. The corresponding numbers refer to the compound numbers provided in Table S3 and S4. The heat maps reveal global differences in exudation patterns and provide evidence that the various treatments (growth medium collected during the exponential or stationary phases, buffered or unbuffered growth medium, 2 bacterial strains) actually led to specific and differentiated exudation patterns. The corresponding individual data (3 biological repeats, and 2 technical repeats per biological repeat, for each "treatment") are all provided in Table S2 and S5. Thus, these data can be used by readers interested in a set of given compounds to check the statistical significance of the differences in relative abundance of these compounds between the different treatments. On the other hand, we believe that introducing supplementary tables for comparing the different "treatments" (and couple of treatments) would lengthen an already abundant iconography. Furthermore, this supplemental iconography could also be considered as not necessary, to our point of view, since the conclusion that has been drawn from these analyses in the revised manuscript, that the different treatments actually resulted in specific and differentiated exudation patterns, is already supported by these heat map figures.
Comment 1.8.
Discussion: Please do not use the too much shorter paragraphs.
Answer and changes in the manuscript.
In the Discussion of the revised manuscript, including in the new Conclusion section (§ 3.4), a single paragraph would appear to be short (6 lines). For sake of clarity, we believe that the corresponding text should be left as an independent paragraph.
Comment 1.9.
Conclusion: Avoid using the references, general statements, focus on the main results and their implications.
Answer and changes in the manuscript.
These comments have been taken into account in the revised manuscript. See also below, our answer to Comment 1.10. A single reference has been maintained in the Conclusion section, and we believe that the corresponding work/article has to be cited at this place.
Comment 1.10.
The most important point: the authors should explain the specific pathway in this study. It is suggested to make the schematic representation of that pathway and the overall study. In addition, this pathway should be mentioned in abstract and conclusion, too.
Answer and changes in the manuscript.
These comments have been taken into account in the revised manuscript. See also our answer to the comment 1.3. A new Figure has been added to the manuscript (Figure 8), which provides a synoptic view of the diverse pathways and mechanisms that the exuded compounds could trigger, underlying the beneficial interactions with the plant. New sentences have also been added to the Conclusion section (§ 3.4) of the revised manuscript. Finally, in the revised manuscript, the last sentence of the Abstract has been rewritten accordingly: "Thus, the methodological processes we have developed to collect and analyze bacterial exudates have allowed to reveal that PGPR constitutively exude a highly complex set of metabolites, which is likely to allow numerous mechanisms to simultaneously contribute to plant growth promotion, and thereby also to broaden the spectrum of plant genotypes (species and accessions/cultivars) with which beneficial interactions can occur".
We sincerely thank both Reviewers for their constructive comments, which have allowed us to improve the manuscript.

Reviewer 2 Report
Zhour et al have provided a study on investigating the effects of diazotrophic bacterial exudates isolated from the rhizosphere of wild wheat on root development in the elite cultivar and to analyze its exo-metabolomes and exo-proteomes. The necessary information is provided in the manuscript. However, there are few points that if considered will increase the value of the manuscript and may be readability.
-Does the manuscript possess a novelty and a future prospect? If yes, it will be good to add the novelty and future prospect in the abstract and introduction.
-One of the main queries is why the bacteria from wild wheat ancestor have been used for the study? Is there any relevance of wild wheat here? If yes, I believe that it will be better adding a common paragraph on wild wheat ancestors in the introduction. For adding information on wild wheat, see https://doi.org/10.3389/fpls.2021.736614.
-Please give the name of the T. dicoccoides genotype used in the study.
-Table 1 is added in the form of a figure. Please correct it.
- Tables and Figure captions should be self-explanatory. Please elaborate on them by mentioning the organism name, treatments etc
I do believe that the manuscript can be accepted once the authors address the mentioned points and enrich the manuscript with crucial information.
Author Response
REVIEWER 2
Comment 2.1.
Does the manuscript possess a novelty and a future prospect? If yes, it will be good to add the novelty and future prospect in the abstract and introduction.
Answer and changes in the manuscript.
The manuscript actually provides novel information. Indeed, for example, no analysis of metabolomes and proteomes exuded by PGPR species had been reported so far to our knowledge. The novelty of such results has been more explicitly underlined in the revised manuscript. These data have revealed that a highly complex and diverse set of compounds is constitutively exuded by the PGPR strains we have characterized. A new Figure (Figure 8) has been introduced in the manuscript in order to provide a synoptic view of the different classes of compounds likely to contribute to beneficial interactions with the plant. The conclusion section (new paragraph 3.4) has been rewritten and lengthened, in particular to discuss this Figure and the conclusions and hypotheses it allows to draw, and to give clues about the perspectives that the work, including the methodological developments, has opened. The abstract has also been partly rewritten to better underline the "take-home message" of the paper. See also our answers to the comments 1.3 and 1.10 of Reviewer 1.
Comment 2.2.
One of the main queries is why the bacteria from wild wheat ancestor have been used for the study? Is there any relevance of wild wheat here? If yes, I believe that it will be better adding a common paragraph on wild wheat ancestors in the introduction. For adding information on wild wheat, see https://doi.org/10.3389/fpls.2021.736614.
Please give the name of the T. dicoccoides genotype used in the study.
Answer and changes in the manuscript.
In the introduction section (lines 116-125), new sentences (and new references) have been introduced to explain the interest of selecting bacteria from wild wheat ancestor rhizosphere (Triticum turgidum dicoccoides, ancestor of tetraploid and hexaploid wheat). Also, the name of the T. dicoccoides genotype (T. turgidum dicoccoides T.t.d-NC-2019) is now indicated in the revised manuscript.
Comment 2.3.
Table 1 is added in the form of a figure. Please correct it.
Answer and changes in the manuscript.
- Thanks. Done.
Comment 2.4.
Tables and Figure captions should be self-explanatory. Please elaborate on them by mentioning the organism name, treatments etc
Answer and changes in the manuscript.
Legends of tables and figures were completed in that way and double-checked.
We sincerely thank both Reviewers for their constructive comments, which have allowed us to improve the manuscript.
Round 2
Reviewer 2 Report
The manuscript can be accepted in its present form.